# Effect of Wounding Intensity on Edible Quality by Regulating Physiological and ROS Metabolism in Fresh-Cut Pumpkins

Wenzhong Hu [1,*,†], Yuge Guan [2,†], Yi Wang [3] and Ning Yuan [3]

1 School of Pharmacy and Food Science, Zhuhai College of Science and Technology, Zhuhai 519041, China
2 School of Food and Health, Zhejiang Agricultural and Forestry University, Hangzhou 311300, China
3 College of Life Science, Dalian Minzu University, Dalian 116600, China
* Correspondence: wenzhongh@sina.com
† These authors contributed equally to this work.

**Abstract:** Fresh-cut pumpkin is favored by consumers for its environmental protection, safety, and convenience at home and abroad. To investigate the effect of different wounding intensities (piece, strip and slice, corresponding to 1.90, 3.53 and 6.29 $m^2\ kg^{-1}$) on the quality of fresh-cut pumpkin, the critical indexes involved in reactive oxygen species (ROS) metabolism, vitamin C-glutathione cycle, phenylpropanoid metabolism and membrane lipid peroxidation were monitored for pumpkin during storage at 4 °C for 6 d. The results showed that with the increase in cutting injury strength, the lightness, whiteness index, respiration rate, ethylene content, lipoxygenase activity and malondialdehyde content of fresh-cut pumpkin increased, while the hardness, sensory quality, appearance and total soluble solid content continuously decreased. The quality deterioration was the most severe in the slice group, while a higher sensory quality was maintained in the piece after 6 d of storage. However, the activity of phenylalanine ammonia-lyase increased and then contributed to the synthesis of the phenolic compound, which resulted in enhancements of 79.13%, 29.47% and 16.14% in piece, strip and slice, respectively. Meanwhile, cutting enhanced the activity of antioxidant enzymes including ascorbate peroxidase, glutathione reductase, superoxide dismutase and catalase, resulting in the enhancement of antioxidant activity in fresh-cut pumpkin. The collected results showed that the wounding intensities have an obvious influence on the quality by regulating physiological and ROS metabolism.

**Keywords:** fresh-cut pumpkins; wounding intensities; reactive oxygen species; edible quality

## 1. Introduction

Pumpkin (*Cucurbita moschata* Duch.) is one of the most common vegetables in the life of consumers, and it is favored by worldwide consumers due to its prominent yellow color and nutritional and functional compounds, such as anthocyanins, phenolic acid derivatives, flavonoids and vitamins [1]. Fresh pumpkin has a large shape and is not convenient for consumers to eat, so the market sale of pumpkin is greatly limited, and the pumpkin product subjected to fresh-cutting treatment can solve the problem. Fresh-cut pumpkins have the characteristics of convenience, safety, nutrition and freshness [2]. Simultaneously, the skin residue, core and other waste material produced via fresh-cut processing can also be recycled and reused for animal feed, which has the characteristics of reducing urban domestic waste and environmental protection [3,4]. In recent years, fresh-cut pumpkins have been increasingly favored by consumers due to their convenient characteristics of ready-to-eat, ready-to-use and ready-to-cook [5,6]. However, cutting causes pumpkins to suffer irreparable mechanical damage and lose their protective tissue, which may induce quality deterioration reactions [7].

In general, when fruits and vegetables are in fresh-cutting processing, a systematic responsive mechanism would be activated. Previous studies have reported that wounding

stress could stimulate the production of reactive oxygen species (ROS) [2]. A study on fresh-cut carrots suggested that ROS may increase the resistance of fresh-cut products by inducing the synthesis and accumulation of secondary metabolites, including polyphenols and flavonoids [8,9]. Chai et al. [10] found that excessive ROS accumulation will break the balance of ROS metabolism in plants, increase the burden of scavenging free radicals and lead to browning and membrane lipid peroxidation, which further reduces the edible quality of fresh-cut products. Therefore, it is essential to regulate ROS metabolism aimed to maintain the quality of fresh-cut products.

It has been reported that cutting styles corresponding to different wounding intensity have an impact on the production of ROS signal in fresh-cut products, and then contribute to the difference in enzymatic reaction and the phenylpropanoid metabolic pathway [11–13]. Generally, during the processing of fresh-cut pumpkins, different cutting methods subjected to piece, strip and slice are necessary for the food industry or at home. However, there is still lacking scientific certification and systematic research on whether the different wounding intensities affect the quality of fresh-cut pumpkins. Moreover, whether ROSs are involved in wound-induced quality change in fresh-cut pumpkins is still unknown. Therefore, three different cutting methods (piece, strip and slice) on the physiological metabolism and quality of fresh-cut pumpkins were evaluated in this study to select the optimal wounding intensities of production and consumption. Meanwhile, the regulation mechanism of ROS on edible quality change of fresh-cut pumpkins was explored, which further provides a certain theoretical basis for fresh-cut pumpkin during processing and storage.

## 2. Materials and Methods

### 2.1. Sample Preparation and Treatment

Our study was conducted from 10 March 2022 to 16 June 2022 in the greenhouse at the Dalian Shengyuan Farmer, PR China (39°15′58.72″ N, 121°55′41.41″ E). Sixty seeds for the Luli pumpkin were sowed on 10 March 2022 in a greenhouse in plastic seedling trays containing potting mix and starter fertilizer in Shengyuan farm, located in Jinzhou District, Dalian, China. At two weeks old, the seedlings of each germplasm line were transplanted on 26 March 2022 into a single plot in the field. The mature tropical pumpkins were harvested at mature green stage (65 days after flowering).

The pumpkin (*Cucurbita moschata*, Duch.) variety named "Luli" was harvested in July 2022. Fresh pumpkins were sorted to ensure uniformity of size, color and ripeness and were maintained at 4 °C before processing. Whole samples were thoroughly washed with tap water and then soaked in sodium hypochlorite (0.2 mL L$^{-1}$) solution for 2 min for sterilization. After being rinsed twice with water, the selected pumpkins were cut into piece, strip and slice. The wounded surface diameter of piece, strip and slice was manually measured, and then wounding intensities were calculated and found to be 1.90, 3.53 and 6.29 m$^2$ kg$^{-1}$, respectively. Fresh-cut pumpkins (120 g) were loaded into a polypropylene container and packaged with polyethylene films (Miuge Chemical Commodities Science and Technology Co., Ltd., Hangzhou, China), then stored at 4 °C. The whole pumpkin was used as the control in this experiment; all samples were collected at 0, 1, 2, 3, 4, 5 and 6 d. In this experiment, 6 of the samples were taken for analysis of lightness, whiteness index, respiration rate and ethylene content, and the other samples were frozen with liquid nitrogen and stored at −80 °C for further analysis.

### 2.2. Lightness, Whiteness Index and Appearance Assay

Tristimulus color values including lightness (*L*\*), redness (*a*\*) and yellowness (*b*\*) of fresh-cut and control samples were determined by using a HunterLab MiniScan@XE plus colorimeter (Hunter Associates Laboratory, Reston, VA, USA); the white and black plank was used as the calibrate standard in this work. For fresh-cut pumpkins, three samples from each group were chosen to measure surface color, taking an average of three surface measurements per sample.

The whiteness index (WI) value was estimated using the formula as shown below, WI $= 100 - [(100 - L^*)^2 + (a^*)^2 + (b^*)^2]^{1/2}$ [14], where $L^*$, $a^*$ and $b^*$ are the color values of lightness, redness and yellowness of samples. The change in the appearance of fresh-cut pumpkins during storage was recorded by obtaining pictures.

### 2.3. Respiration Rate and Ethylene Content Assay

The respiration rate and ethylene content were determined via a gas analyzer (F-940 STORE II, Felix, USA) according to the method reported by Zhou et al. [15]. The respiration rate of fresh-cut pumpkin was calculated as follows:

$$\text{Respiration rate} = \frac{(C_1 - C_2)/100 * V}{m * t} * 1.96$$

where $C_1$ represents the volume fraction of $CO_2$ in the container (%); $C_2$ represents the volume fraction of $CO_2$ in the air (%); V is the volume of the container (mL); m represents the weight of the sample (kg); t represents time (h); 1.96 is the density of $CO_2$ (g $L^{-1}$).

The ethylene content ($\mu L\ kg^{-1}\ h^{-1}$) of the sample was calculated as follows:

$$\text{Ethylene content} = \frac{(C_1 - C_2) * V}{m * t * 1000}$$

where $C_1$ represents the volume fraction of ethylene in the container (%); $C_2$ represents the volume fraction of ethylene in the air (%); V is the volume of the container (mL); m represents the weight of the sample (kg); t represents time (h).

### 2.4. $O_2^{-\bullet}$ and $H_2O_2$ Content Assay

The $O_2^{-\bullet}$ and $H_2O_2$ content were measured by using commercially available kits (Shanghai Yuanye Biotechnology Co., Ltd., Shanghai, China). The absorbance of $O_2^{-\bullet}$ and $H_2O_2$ reaction system was monitored at 530 nm and 412 nm, respectively, and then the result was expressed as mmol $kg^{-1}$.

### 2.5. Malondialdehyde (MDA) Content and Lipoxygenase (LOX) Activity Assay

The MDA content was determined by using the color reaction between MDA and thiobarbituric acid, the measurement method was in accordance with the plant kit (www.cominbio.com, accessed on 15 September 2022) and the result was expressed in mmol $kg^{-1}$.

The MDA content was determined by using the color reaction between MDA and thiobarbituric acid by using the plant kit (Coming Biotechnology Co., Ltd., Suzhou, China). The MDA content was determined according to the instructions of the manufacturer [16], and the result was expressed mmol $kg^{-1}$.

LOX catalyzes the oxidation of linoleic acid, the oxidation product has a characteristic absorption peak at 280 nm, and the LOX activity was calculated by measuring the increase rate at 280 nm absorbance. The determination method was in accordance with the plant kit (Coming Biotechnology Co., Ltd., Suzhou, China) [17]; the result was expressed in U $kg^{-1}$, where $U_{LOX} = 0.01 \times \Delta A_{280}$ nm per min based on fresh tissue weight.

### 2.6. Lignin Content and Phenylalanine Ammonia-Lyase (PAL) Activity Assay

The samples were dried to constant weight at 80 °C, and then crushed and sieved through a 40 mesh sieve, which was prepared for the extraction of lignin. The lignin content was measured with plant kits (Coming Biotechnology Co., Ltd., Suzhou, China), the method was in accordance with the instructions of the manufacturer [18] and the result was expressed as g $kg^{-1}$ based on fresh weight.

The PAL activity was assayed as described by Guan et al. [6]. Frozen samples (1.0 g) were extracted in 5 mL of ice-cold sodium borate buffer consisting of 5 mmol $L^{-1}$ β-mercaptoethanol. The 2.5 mL reaction system comprised borate buffer (2 mL), L-phenylalanine (1 mL; 20 mmol $L^{-1}$) and extracted supernatant (0.5 mL). The absorbance

change was measured at 290 nm and one U of PAL was described as the amount of enzyme that causes a 0.01 absorbance change per minute.

### 2.7. Antioxidant Activity Assay

The antioxidant activity was assessed by analyzing the hydroxyl radical (OH$^\bullet$), 2,2′-azinobis-3-ethylbenzthiazoline-6-sulphonate (ABTS) scavenging ability and ferric reducing antioxidant power (FRAP) by using plant kits (Nanjing Jiancheng Bioengineering Institute Co., Ltd., Nanjing, China). The OH$^\bullet$, ABTS and FRAP scavenging ability was determined according to the instructions of the manufacturer; the specific steps were shown at the company home page [19–21]. The result of OH$^\bullet$ and ABTS scavenging ability was expressed as the formula: scavenging ability (%) = $100 \times (A_{control} - A_{sample})/A_{control}$, where $A_{control}$ and $A_{sample}$ are the absorbances of the control and of the sample, respectively. The FRAP result was expressed as mmol kg$^{-1}$ based on a Trolox calibration curve.

### 2.8. Antioxidant Enzyme Activity Assay

The pumpkin samples were extracted using ice-cold sodium phosphate buffer (0.2 mol L$^{-1}$, pH 6.4) and then centrifuged at $12,000 \times g$ for 30 min at 4 °C, the collected supernatant was prepared to measure polyphenol oxidase (PPO), peroxidase (POD) and catalase (CAT) activity. The PPO, POD and CAT activity was determined as reported by Tian et al. [22] and Chen et al. [23], and the results were expressed as U kg$^{-1}$.

The activity of superoxide dismutase (SOD), ascorbate peroxidase (APX) and glutathione reductase (GR) was measured using kits (Comin Biotechnology Co., Ltd., Suzhou, China). The SOD activity was determined according to the instructions of the manufacturer [24–26]. The results were expressed as U kg$^{-1}$, where $U_{SOD} = 0.01 \times \Delta_{450}$ nm per min, $U_{APX} = 0.01 \times \Delta_{290}$ nm per min and $U_{GR} = 0.01 \times \Delta_{340}$ nm per min.

### 2.9. Antioxidant Compounds and Total Soluble Solid (TSS) Content Assay

The antioxidant compounds of pumpkin including phenols, flavonoids, anthocyanin, Vitamin C (Vc) and glutathione (GSH) content affected by wounding intensities were evaluated in this study. For total phenol content evaluation, the frozen pumpkin sample (5 g) was mixed with 20 mL 80% ethanol and extracted in the dark at 40 °C for 40 min and the obtained supernatant was prepared to determine the value, expressed as mg kg$^{-1}$ based on the gallic acid standard curve [6].

The method of flavonoid content measurement was in accordance with Santana-Galvez [7]. Fresh tissues (0.5 g) were mixed with 5 mL of 99% HCl-methanol for 20 min at 4 °C. The absorbance changes of the reaction system were measured at 325 nm to determine total flavonoid content based on the rutin equivalent standard curve (g kg$^{-1}$).

Then, the content of anthocyanin, Vc and GSH was determined by using plant kits (Solarbio Science & Technology Co, Ltd., Beijing, China) in accordance with the manufacturer's instructions. The absorbance of anthocyanin, Vc and GSH was measured at 500 nm, 420 nm and 412 nm, respectively. Anthocyanin, Vc and GSH content are expressed as g kg$^{-1}$, mg kg$^{-1}$ and mmol kg$^{-1}$, respectively.

TSS content was assayed according to the process of Li et al. [8] by using a hand-held digital refractometer (PAL-1, ATAGO, Tokyo, Japan), expressed as %.

### 2.10. Statistical Analysis

All the experiments were performed in triplicates, and data were presented as the mean ± SD (standard deviation). Orignpro 2021 software (OriginLab., Northampton, MA, USA) was used for the drawing of figures. One-way analysis of variance (ANOVA) and least significant difference (LSD) tests via SPSS software, version 20 (IBM Corp., Armonk, NY, USA) were used to differentiate mean values at the $p < 0.05$ level.

## 3. Results and Discussion

### 3.1. Colorimetric Values and Appearance

The colorimetric value evaluation involved the important quality parameters of food [27]. In order to quickly and intuitively illuminate the visual color change of fresh-cut pumpkin during storage at 4 °C, we measured the lightness ($L^*$) as shown in Figure 1a. The result revealed that the $L^*$ of cutting treatment groups increased, and the rising trend of color difference $L^*$ value in a whole pumpkin was significantly lower than that in other fresh-cutting treatment groups. On storage for 6 d, the $L^*$ of whole, piece, strip and slice was 74.9, 78.6, 79.2 and 81.07, respectively, which indicated that the degree of $L^*$ level was strip > piece > slice > whole (Figure 1a); the above results show that with the increase in the wounding intensities, the color difference $L^*$ value of the fresh-cut pumpkin has an increasing trend. Furthermore, the whiteness index (WI) was used to indicate the degree of blanching of fresh-cut pumpkins during storage. According to Figure 1b, the WI was increased during the whole storage period. On storage for 6 d, the WI of the slice was reached at the maximum value (56.79), which was 1.69 times higher than the control group. The result of the sensory evaluation (Figure 1c) also demonstrated that the score of appearance, color, taste, texture and overall appearance in slice was lower than strip and piece, which indicated the piece pumpkin had the best visible quality.

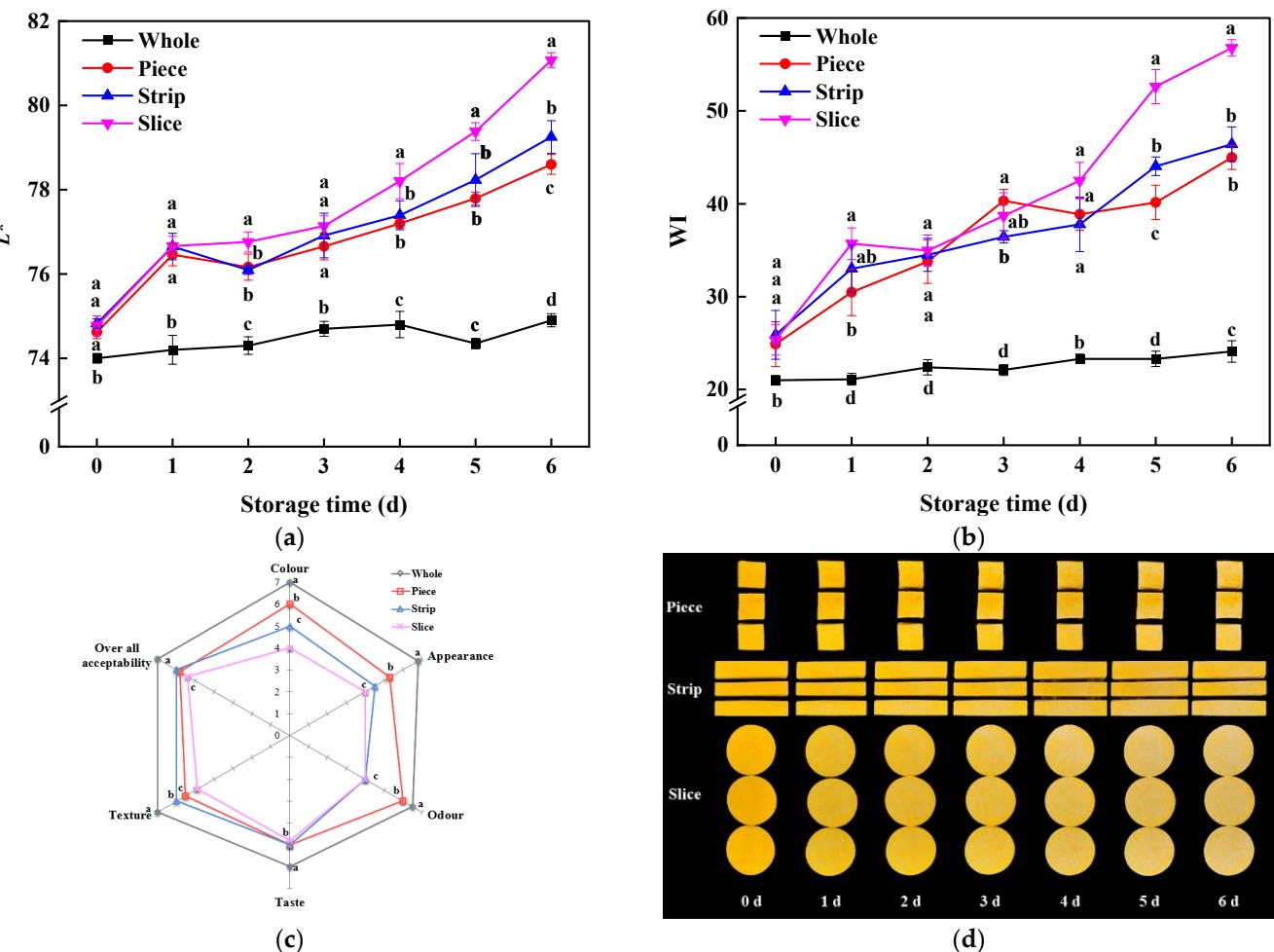

**Figure 1.** Effect of wounding intensities on lightness ($L^*$) (**a**), whiteness index (WI) (**b**), sensory evaluation (**c**) and appearance (**d**) of fresh-cut pumpkin during storage at 4 °C for 6 days. Vertical bars represent the standard deviation of the mean (n = 3). Different letters indicate significant differences between the groups ($p < 0.05$).

The phenomenon of whitening on the cut surface of pumpkin was easy to observe based on the picture of the appearance (Figure 1d); it was also found in fresh-cut carrots that the surface is apt to white blush [28]. The reason for this may be that the color compounds of fresh-cutting processed fruits and vegetables changed during storage due to incomplete inactivation of enzymes, which would result in the white blush on the surface [29]. However, this result was opposite to the quality change of fresh-cut potatoes [11] and apples [30] where the degree of browning phenomenon was deep. The above results also showed that the color changes were significantly different in fresh fruits and vegetables with different biological characteristics after being sliced.

### 3.2. Weight Loss and Total Soluble Solid (TSS) Content

Weight loss was increased significantly for fresh-cut pumpkin (Figure 2a). Among these groups, the weight loss of sliced pumpkin was highest (1.49%), followed by strip and piece. As for TSS content in fresh-cut pumpkin, it was affected by wounding intensities, significantly decreasing during the whole storage period (Table 1), yet it surprisingly remained almost constant in whole pumpkin during storage. Contrary to the weight loss, the TSS content of the slice group was lowest; it decreased by 32.49% at 7 d compared with the original values. A significant negative correlation (R = −0.885) was observed between weight loss and TSS content, which indicated that fresh-cutting operations easily cause water loss and further resulted in TSS content change in fresh-cut pumpkin [31,32].

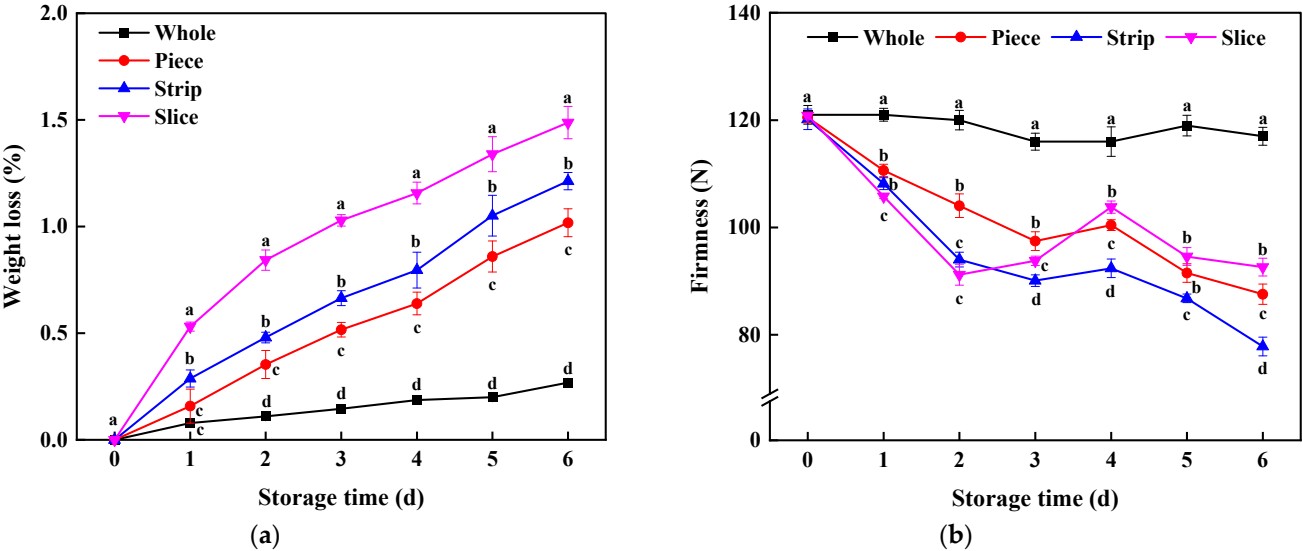

**Figure 2.** Effect of wounding intensities on weight loss (**a**) and firmness (**b**) of fresh-cut pumpkin during storage at 4 °C for 6 days. Vertical bars represent the standard deviation of the mean (n = 3). Different letters indicate significant differences between the groups (*p* < 0.05).

Firmness is an important apparent property of fresh-cut products [33]; just after fresh-cutting, the firmness measured on pumpkin was 120.16 N, then it decreased sharply (Figure 2b). Notably, the firmness of piece, strip and slice showed a slight significant increase at 3–4 d. Probably, the cutting operation with wounding stress applied during the experiment resulted in wound healing and then provided an essential barrier for a compact superficial surface layer, which contributed to surface hardening [1].

**Table 1.** Influence of wounding intensities on the quality of fresh-cut pumpkin.

| Index | Treatment | 0 d | 1 d | 2 d | 3 d | 4 d | 5 d | 6 d |
|---|---|---|---|---|---|---|---|---|
| TSS (%) | Whole | 16.00 ± 0.54 [a] | 15.70 ± 0.20 [a] | 15.30 ± 0.50 [a] | 15.61 ± 0.06 [a] | 15.13 ± 0.44 [a] | 15.52 ± 0.11 [a] | 16.03 ± 0.07 [a] |
| | Piece | 15.65 ± 0.39 [a] | 13.81 ± 0.11 [c] | 12.45 ± 0.39 [b] | 14.57 ± 0.34 [b] | 14.23 ± 0.37 [b] | 12.97 ± 0.11 [b] | 12.54 ± 0.29 [b] |
| | Strip | 15.87 ± 0.13 [a] | 14.62 ± 0.40 [b] | 11.67 ± 0.19 [c] | 12.32 ± 0.33 [c] | 13.12 ± 0.29 [c] | 11.97 ± 0.13 [c] | 11.87 ± 0.15 [c] |
| | Slice | 15.45 ± 0.21 [a] | 13.52 ± 0.16 [d] | 11.55 ± 0.37 [c] | 10.75 ± 0.05 [d] | 12.22 ± 0.22 [d] | 11.07 ± 0.07 [d] | 10.43 ± 0.11 [d] |
| Total phenols content (g kg⁻¹) | Whole | 10.74 ± 0.42 [c] | 10.19 ± 0.25 [c] | 10.04 ± 0.11 [c] | 9.92 ± 0.23 [d] | 10.19 ± 0.44 [d] | 10.35 ± 0.22 [d] | 10.53 ± 0.22 [c] |
| | Piece | 11.16 ± 0.26 [b] | 12.01 ± 0.19 [b] | 13.47 ± 0.21 [a] | 12.74 ± 0.26 [c] | 12.70 ± 0.38 [c] | 12.02 ± 0.29 [c] | 11.07 ± 0.32 [b] |
| | Strip | 11.32 ± 0.35 [b] | 12.32 ± 0.20 [ab] | 12.41 ± 0.23 [b] | 13.52 ± 0.21 [b] | 14.58 ± 0.33 [b] | 13.40 ± 0.22 [b] | 11.61 ± 0.23 [b] |
| | Slice | 12.07 ± 0.28 [a] | 12.71 ± 0.31 [a] | 13.92 ± 0.29 [a] | 15.74 ± 0.04 [a] | 16.30 ± 0.44 [a] | 18.54 ± 0.24 [a] | 12.49 ± 0.34 [a] |
| Flavonoids content (g kg⁻¹) | Whole | 5.95 ± 0.17 [a] | 5.46 ± 0.21 [c] | 5.57 ± 0.35 [c] | 5.89 ± 0.16 [d] | 5.45 ± 0.11 [d] | 5.49 ± 0.31 [c] | 5.58 ± 0.15 [b] |
| | Piece | 5.83 ± 0.19 [a] | 7.60 ± 0.25 [a] | 7.58 ± 0.38 [b] | 8.40 ± 0.19 [b] | 8.97 ± 0.28 [b] | 7.89 ± 0.18 [b] | 7.08 ± 0.19 [a] |
| | Strip | 5.63 ± 0.44 [a] | 6.66 ± 0.12 [b] | 7.37 ± 0.16 [b] | 7.69 ± 0.14 [c] | 7.79 ± 0.16 [c] | 8.21 ± 0.11 [b] | 7.20 ± 0.16 [a] |
| | Slice | 5.98 ± 0.55 [a] | 7.65 ± 0.24 [a] | 8.46 ± 0.18 [a] | 9.10 ± 0.41 [a] | 11.04 ± 0.28 [a] | 8.57 ± 0.07 [a] | 7.44 ± 0.26 [a] |
| Anthocyanin content (g kg⁻¹) | Whole | 0.11 ± 0.04 [a] | 0.13 ± 0.01 [c] | 0.13 ± 0.03 [c] | 0.11 ± 0.01 [c] | 0.16 ± 0.01 [d] | 0.13 ± 0.04 [c] | 0.15 ± 0.01 [c] |
| | Piece | 0.17 ± 0.03 [a] | 0.19 ± 0.02 [ab] | 0.21 ± 0.02 [b] | 0.24 ± 0.01 [b] | 0.55 ± 0.02 [b] | 0.24 ± 0.03 [b] | 0.22 ± 0.02 [b] |
| | Strip | 0.15 ± 0.01 [a] | 0.18 ± 0.02 [b] | 0.20 ± 0.03 [ab] | 0.28 ± 0.02 [a] | 0.35 ± 0.01 [c] | 0.45 ± 0.03 [a] | 0.21 ± 0.02 [b] |
| | Slice | 0.14 ± 0.02 [a] | 0.24 ± 0.03 [a] | 0.26 ± 0.03 [a] | 0.31 ± 0.01 [a] | 0.72 ± 0.01 [a] | 0.50 ± 0.02 [a] | 0.28 ± 0.03 [a] |
| Vitamin C (mg kg⁻¹) | Whole | 347.94 ± 1.55 [a] | 350.30 ± 11.28 [a] | 348.57 ± 5.77 [a] | 333.85 ± 1.46 [a] | 337.62 ± 11.41 [a] | 334.68 ± 2.08 [a] | 340.39 ± 1.42 [a] |
| | Piece | 342.21 ± 10.40 [a] | 345.30 ± 10.00 [a] | 315.64 ± 5.13 [b] | 315.76 ± 8.62 [b] | 299.77 ± 7.64 [b] | 287.70 ± 3.54 [b] | 245.30 ± 2.03 [b] |
| | Strip | 344.06 ± 1.85 [a] | 275.20 ± 7.61 [c] | 230.15 ± 2.76 [c] | 218.55 ± 7.36 [c] | 222.85 ± 7.80 [c] | 231.59 ± 1.92 [c] | 228.97 ± 5.30 [c] |
| | Slice | 325.56 ± 3.11 [b] | 322.27 ± 2.96 [b] | 222.69 ± 5.04 [c] | 165.27 ± 7.23 [d] | 159.38 ± 13.80 [d] | 179.79 ± 3.88 [d] | 193.57 ± 7.68 [d] |
| Glutathione (mmol kg⁻¹) | Whole | 0.23 ± 0.004 [a] | 0.20 ± 0.015 [c] | 0.25 ± 0.011 [d] | 0.27 ± 0.007 [d] | 0.24 ± 0.012 [d] | 0.25 ± 0.015 [d] | 0.26 ± 0.015 [d] |
| | Piece | 0.20 ± 0.006 [c] | 0.15 ± 0.007 [d] | 0.29 ± 0.005 [c] | 0.52 ± 0.015 [c] | 0.52 ± 0.006 [c] | 0.35 ± 0.007 [c] | 0.43 ± 0.014 [c] |
| | Strip | 0.22 ± 0.004 [b] | 0.24 ± 0.011 [b] | 0.41 ± 0.004 [b] | 0.68 ± 0.002 [b] | 0.75 ± 0.016 [b] | 0.58 ± 0.009 [b] | 0.58 ± 0.005 [b] |
| | Slice | 0.23 ± 0.017 [ab] | 0.33 ± 0.003 [a] | 0.54 ± 0.004 [a] | 0.88 ± 0.006 [a] | 0.80 ± 0.016 [a] | 0.77 ± 0.006 [a] | 0.87 ± 0.014 [a] |

Note: Influence of wounding intensities on TSS, total phenols, flavonoids, anthocyanin, vitamin C and glutathione content of fresh-cut pumpkin during 6 d of storage at 4 °C. Data are expressed as the mean ± SD. Values with different letters were significantly different at $p < 0.05$. Lowercase letters represented significant difference among treatment factors.

### 3.3. Effect of Wounding Intensities on Physiological Metabolism of Fresh-Cut Pumpkins

#### 3.3.1. Respiration Rate and Ethylene Content

Respiratory rate is an important condition for maintaining the freshness and shelf life of produce [34]. In this study, the trend for respiration rate of slice, strip and piece was increased first and reached the peak value on the 3rd day, and increased 1.83, 2.26, and 2.55 times compared with the whole sample (Figure 3a), revealing that the cutting operation treatment caused wounding stress in fresh tissues and further resulted in triggering respiratory metabolism of fresh-cut pumpkin [35]. Similar results were found in fresh-cut potatoes [15], apples [23] and cabbage [27]; all these results indicate that cutting causes fruit and vegetables to suffer irreparable mechanical damage and lose their protective tissue.

As for ethylene, an important plant signal molecule, it was inspired by wounding stress in pumpkins. The highest ethylene content in slice, strip and piece was 14.32, 5.72 and 1.29-fold higher than the control, respectively. During the whole storage period, the enhancement of ethylene content was as follows: slice > strip > piece ($p < 0.05$, Figure 3b). The result of a significant positive correlation (R = 0.636) between ethylene and respiration rate confirmed that ethylene plays a crucial role in regulating the respiratory process during the processing of fresh-cut pumpkin, and this may be an important reason that directly affected the pumpkin tissue senescence [36,37].

#### 3.3.2. Phenylpropane Metabolism-Related Parameters

When plants are subjected to mechanical damage and stress, the comprehensive defense system would be induced [2,38]. Among these, the phenylpropanoid pathway plays a critical role in the process of resisting wounding stress, by activating the production of beneficial secondary metabolites including phenolics, flavonoids and lignin [39]. The previous study of fresh-cut cabbage [27], potato [11], carrot [40] and pitaya [10] illustrated that wounding stress enhances phenolic and flavonoid content, whereas the effects of different wounding intensities on these secondary metabolites in pumpkins are scarcely reported. In this study, we found that the wounding stress increased the total phenol contents in pumpkin (Table 1). Total phenol content of slice, strip and piece increased

by 79.13%, 29.47% and 16.14%, respectively, at 5 d compared with the whole pumpkin (Table 1). As one of the important biological compounds in plants, flavonoids have been reported to contain important antioxidant substances [41,42]. In this study, we found that the flavonoid content increased first, at 4 d; the total flavonoid content of slice, strip and piece pumpkins increased 1.03-fold, 0.43-fold and 0.65-fold, respectively, compared with the control group (Table 1). It can be clearly seen from the above results that total phenol and flavonoid content was higher with the increasing wounding intensity, similar results were found in fresh-cut broccoli [6] and potato [11].

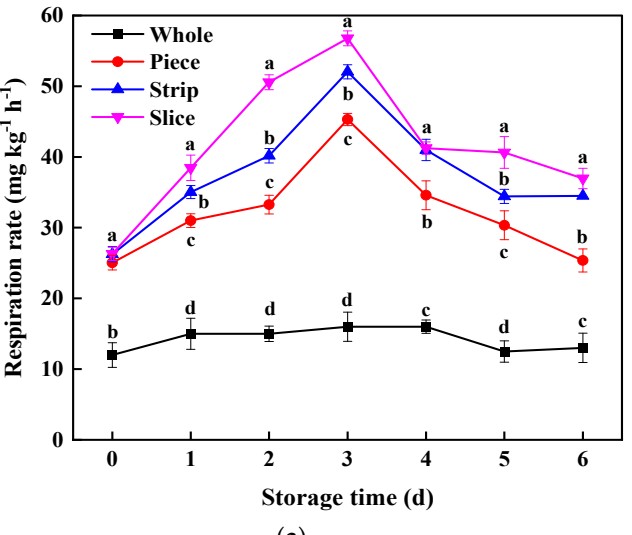
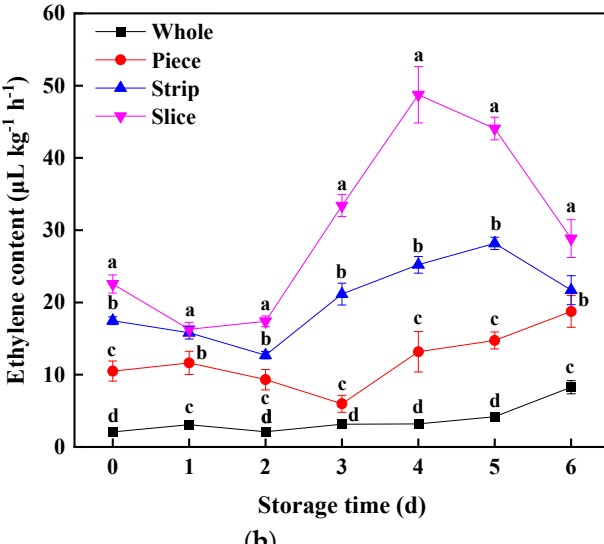

(**a**)  (**b**)

**Figure 3.** Effect of wounding intensities on respiration rate (**a**) and ethylene content (**b**) of fresh-cut pumpkin during storage at 4 °C for 6 days. Vertical bars represent the standard deviation of the mean (n = 3). Different letters indicate significant differences between the groups ($p < 0.05$).

Lignin is widely regarded as an essential secondary metabolite in agricultural products and it plays crucial role in repairing wounding and maintaining a solid shape as well as alleviating oxidative damage and preventing the invasion of pathogenic bacteria [2,43]. The present work revealed that fresh-cut pumpkins accumulated higher lignin content (Figure 4a), and the lignin levels of fresh-cut pumpkin were slice > strip > piece, and the total lignin content in slice, strip and piece increased by 164.18%, 134.32% and 68.76% compared with whole pumpkins (Figure 4a), respectively. In the phenylpropanoid pathway, PAL is the critical enzyme in the biosynthesis of phenols [13]. According to Figure 4b, PAL activity of piece, strip and slice pumpkins was all induced by the cutting process; they arrived at their peaks at 2 d, 3 d and 4 d, respectively, and then declined overall afterward. The significant positive correlation results among PAL activity, phenols and lignin content clarified that the fresh-cutting operation increased PAL activity to promote the synthesis of phenolic substances, which further provide sufficient precursor substances to generate lignin.

### 3.3.3. Membrane Lipid Peroxidation-Related Parameters

Generally, MDA is the end product of membrane lipid peroxidation, and it is considered a biomarker in plants [44,45]. As shown in Figure 5a, the MDA content of the fresh-cut pumpkin group fluctuated throughout, and the overall change was significant. Throughout the storage period, the MDA content of the slice, strip and piece was obviously higher than that of the whole group. At the end of the storage time (6 d), the MDA content in slice, strip and piece increased by 88.13%, 80.48% and 56.06% compared with whole pumpkins, respectively. The enhancement in MDA content indicated that wounding stress accelerated the membrane lipid peroxidation in fresh-cut pumpkins.

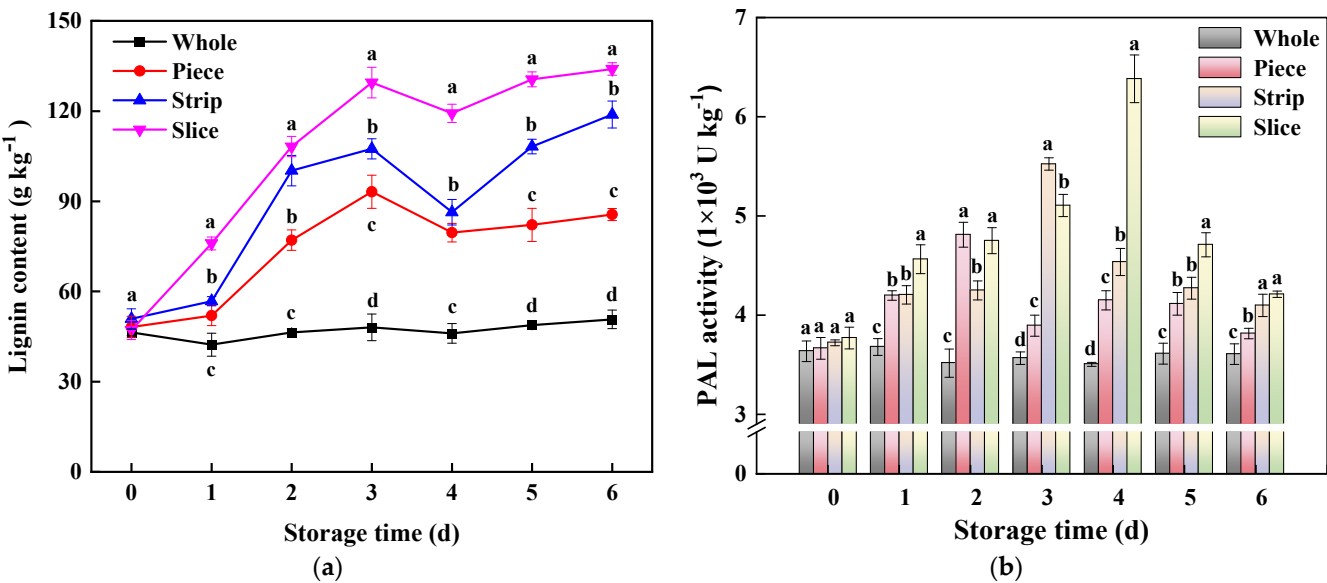

**Figure 4.** Effect of wounding intensities on lignin content (**a**) and phenylalanine ammonia-lyase (PAL) activity (**b**) of fresh-cut pumpkin during storage at 4 °C for 6 days. Vertical bars represent the standard deviation of the mean (n = 3). Different letters indicate significant differences between the groups (*p* < 0.05).

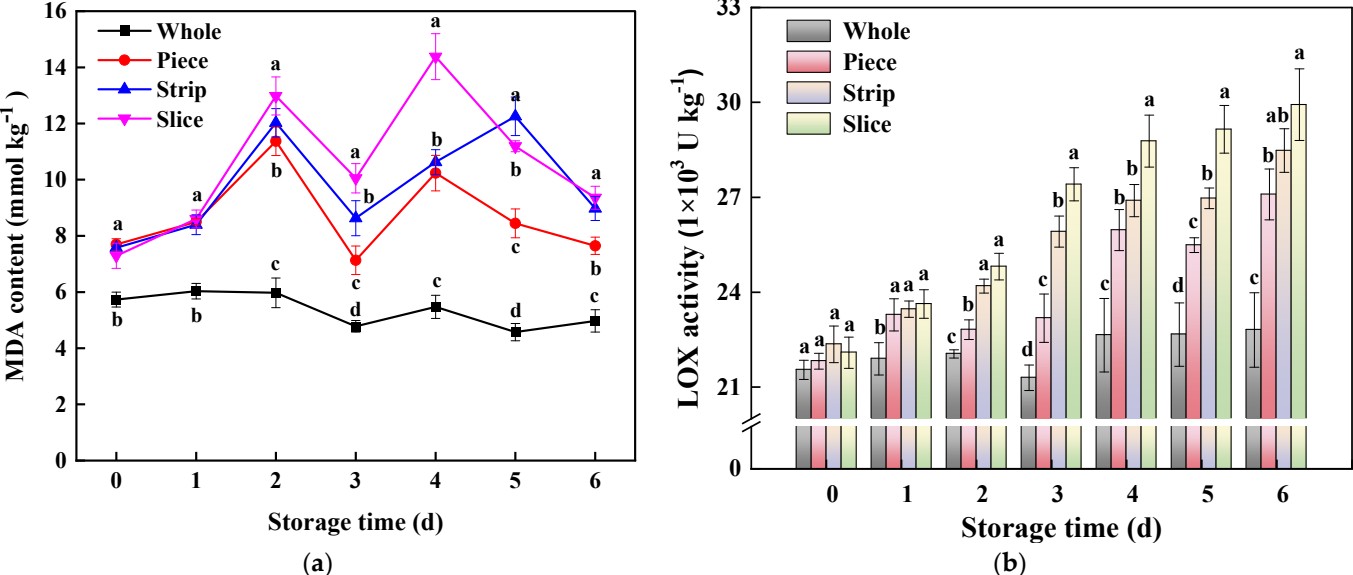

**Figure 5.** Effect of wounding intensities on MDA content (**a**) and LOX activity (**b**) of fresh-cut pumpkin during storage at 4 °C for 6 days. Vertical bars represent the standard deviation of the mean (n = 3). Different letters indicate significant differences between the groups (*p* < 0.05).

LOX is a vital enzyme in membrane lipid metabolism by catalyzing polyunsaturated fatty acids to produce hydrogen peroxide with conjugated double bonds [46]. According to Figure 5b, LOX activity was continuous in the upward trend. At 6 d, the LOX activity in slice, strip and piece increased by 31.22%, 24.91% and 18.82% compared with the whole pumpkin, respectively. In this study, we found that there was a significant positive correlation (R = 0.649) between LOX and MDA; similar results were found in fresh-cut potatoes [11], lotus roots [47], pears [46] and Chinese water chestnuts [48] where LOX could enhanced MDA accumulation through oxidative damage. However, significant negative correlations were found among LOX and TSS content (−0.822) and firmness (−0.772), indicating that fresh-cutting reduced TSS content and firmness of pumpkins by

stimulating LOX enzyme activity. This may be because the enhancement of LOX led to the membrane lipid peroxidation, and then promoted leakage of cellular substances and further contributed to the reduction in TSS and firmness.

### 3.4. Effect of Wounding Intensities on ROS Metabolism of Fresh-Cut Pumpkins

#### 3.4.1. ROS Content

As the key signal molecules in plants, ROS regulate the integration of signal transmission and activation of the wounding stress response network [49]. The major forms of ROS in fruits and vegetables, which vary greatly in their functions and characteristics, mainly include superoxide ($O_2^{-\bullet}$) and hydrogen peroxide ($H_2O_2$) [50]. According to Figure 6a, the $O_2^{-\bullet}$ content in fresh-cut pumpkin was immediately induced by the cutting process; similar results were found in various products, such as apple, which suggests that fresh-cut treatment caused excessive oxidative stress. Moreover, this study revealed that $O_2^{-\bullet}$ content increases with the increase in wounding intensities. This may be because with the wounding intensity increasing, the larger wounded surface is in contact with $O_2$, which speeds up the process of peroxidation and leads to the accumulation of $O_2^{-\bullet}$ [7].

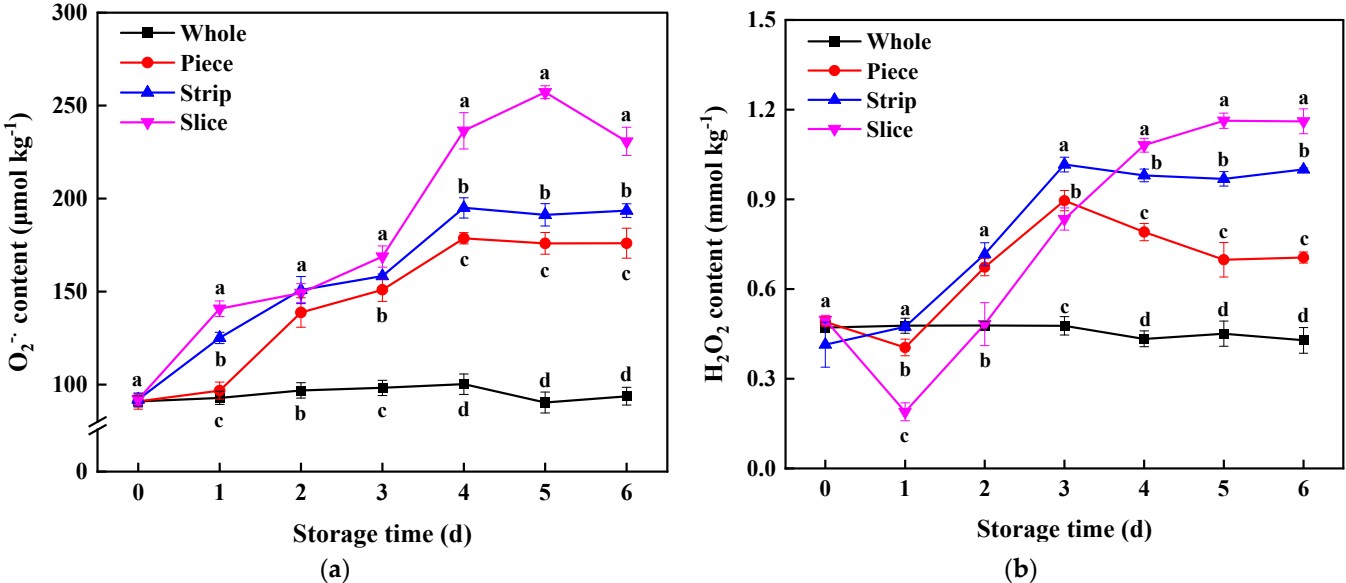

**Figure 6.** Effect of wounding intensities on $O_2^{-\bullet}$ (**a**) and $H_2O_2$ content (**b**) of fresh-cut pumpkin during storage at 4 °C for 6 days. Vertical bars represent the standard deviation of the mean (n = 3). Different letters indicate significant differences between the groups ($p < 0.05$).

Different from the $O_2^{-\bullet}$ change trend, $H_2O_2$ content in slice pumpkin with the highest wounding intensity was lower than strip and piece during the first 3 d storage time (Figure 6b). Whereas $H_2O_2$ content also increased with the enhancement of wounding intensity after 3 d storage, the $H_2O_2$ content in slice, strip and piece groups increased 1.70 times, 1.33 times and 0.65 times, respectively, compared with that of whole pumpkins. There was a significant positive correlation between ROS ($O_2^{-\bullet}$, $H_2O$) and membrane lipid peroxidation-related parameters (LOX, MDA); similar results were obtained from the obvious study that cutting induced the ROS to burst and further caused higher membrane integrity and oxidative stress to accelerate quality deterioration [51,52]. Additionally, a significant positive correlation between ROS ($O_2^{-\bullet}$, $H_2O$) and phenylpropane metabolism-related indexes (PAL, phenols and lignin content) was also found in this study, revealing the fact that ROS involves signaling molecules in fresh-cut pumpkins.

### 3.4.2. Antioxidant Compound Content

In plants, anthocyanin is a natural pigment existing and is given the color of vegetables [53]. As shown in Table 1, anthocyanin content increased in all treatments during the early stage of storage, and then decreased after 5 d of storage time. At 4 d of storage, the anthocyanin content of slice and piece reached the maximum values, which increased 3.50 times and 2.44 times in comparison with that of whole pumpkins, respectively. The anthocyanin content of the strip reached the maximum values at 5 d of storage, which was an increase from $0.15 \, \text{g kg}^{-1}$ to $0.45 \, \text{g kg}^{-1}$. This finding was similar to the previous study on red cabbage that the cutting operation resulted in a remarkable enhancement of anthocyanin [27]. The anthocyanin content decreased at the later storage period, which could be explained by the color fading appearance by the increase in WI values infresh-cut pumpkins.

Vc is derived from fresh fruits and vegetables and is an essential nutrient for the human body [49]. The effects of wounding intensities on the Vc of pumpkin are shown in Table 1. Vc content decreased continuously, and the Vc content of fresh-cut groups was lower than the whole group throughout the storage. At the end of the storage time, the Vc content of slice, strip and piece was decreased by 43.24%, 32.65% and 27.94%, respectively, compared with the whole pumpkin. Clearly, remarkable differences were observed in the Vc content in the three wounding intensities. However, a previous study on fresh-cut broccoli found that the cutting types had no significant effect on Vc content [6]. This may be because of a unique property of pumpkin in which Vc decomposition and oxidation process was relatively sensitive to wounding stress. Furthermore, different from the previous reports that the temperature was 20 °C during the storage of fresh-cut broccoli, 4 °C was applied in this study [6].

GSH was the most important antioxidant sulfhydryl substance involved in the Vc-GSH metabolism; it interacted with Vc to scavenge free radicals in plants [54]. In this study, GSH content increased sharply in fresh-cut pumpkins, which is associated with wounding intensity (Table 1). At 6 d of storage time, the GSH contents of slice, strip and piece increased 2.35, 1.23 and 0.65 times compared with that of the whole group, respectively. A similar phenomenon was also observed in fresh-cut red cabbage [55] and potato [56].

### 3.4.3. ROS Metabolism-Related Enzyme Activity

It is well known that cutting causes fruit and vegetables to suffer irreparable mechanical damage and lose their protective tissue, which may induce ROS accumulation [57]. At the same time, the plant machinery will start its own enzyme system to regulate the balance of ROS to reduce oxidative damage. In general, there were two types of resistance enzymes, oxidase and antioxidant enzymes. Among these, polyphenol oxidase (PPO) and peroxidase (POD) were the most important oxidase enzyme that catalyzes phenolic substrates to quinone with the presence of ROS [58]. According to Figure 7a, the PPO activity of slice, strip and piece increased sharply and reached the maximum at 1 d, which was an of increase 3.91, 2.66, and 1.50 times, respectively, compared with that of the whole pumpkin. A similar result was found in POD activity; it was induced sharply by wounding stress, and POD activity during storage was ranked as follows: slice > strip > piece (Figure 7b). Furthermore, a significant positive correlation appeared between oxidase enzyme (PPO, POD) activity and phenol content, and was also found in PPO, POD activity and MDA content, which confirmed that PPO and POD accelerated the oxidation process of phenols and promoted the oxidation process of membrane lipids in fresh-cut pumpkin. The previous study on fresh-cut apple [23] also addressed that the PPO and POD play a catalytic role in the process of oxidative damage.

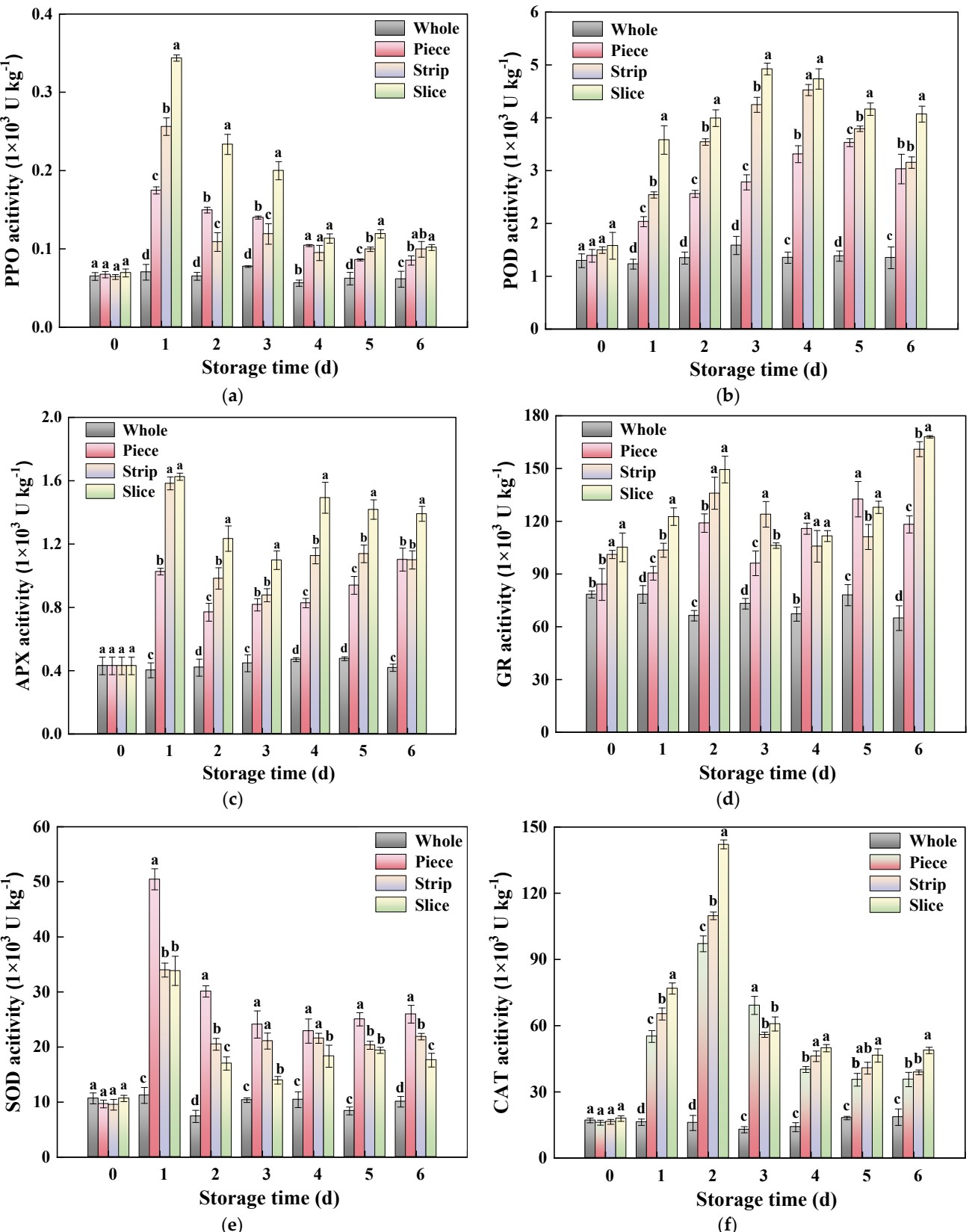

**Figure 7.** Effect of wounding intensities on PPO (**a**), POD (**b**), APX (**c**), GR (**d**), CAT (**e**) and SOD (**f**) activity of fresh-cut pumpkin during storage at 4 °C for 6 days. Vertical bars represent the standard deviation of the mean (n = 3). Different letters indicate significant differences between the groups ($p < 0.05$).

It has been reported that the Vc-GSH cycle was one of the antioxidant enzyme systems, in which ascorbate peroxidase (APX) and glutathione reductase (GR) interacted with antioxidant substances (Vc and GSH) to scavenge excess ROS and alleviate oxidative damage in fresh-cut products [59,60]. In the present work, the fresh-cutting operation induced the APX and GR activity, and the APX and GR activity of cutting groups was higher than whole pumpkin during the whole storage time (Figure 7c,d). It is worth mentioning that APX activity was significantly negatively correlated with Vc content (−0.727), whereas GR activity was significantly positively correlated with GSH content (0.599). Meanwhile, the APX and GR activity was significantly positively correlated with ROS content. The above results revealed that the existence of ROS promoted the process of Vc-GSH cycle by regulating APX and GR activity.

In plants, SOD could catalyze $O_2^{-\bullet}$ into $H_2O_2$, which is then decomposed by CAT [50,61]. As depicted in Figure 7e, the SOD activity of fresh-cut kiwifruits increased substantially and reached the maximum on the first day of storage, then followed by a slight decrease until the end of storage. Different from the other antioxidant enzymes, the order of SOD activity was: slice < strip < piece. Similarly, CAT activity was induced by wounding stress in fresh-cut pumpkin (Figure 7f), and it was enhanced with the higher wounding intensity. At 2 d, the CAT activity in slice, strip and piece was increased by 7.94, 5.90 and 5.11 times compared with whole pumpkin, respectively.

### 3.4.4. Antioxidant Activity

Studies on lettuce [62], cauliflower [63] and carrot [64] have demonstrated that ABTS and FRAP are very important assays for the evaluation of antioxidant activity in vegetables. In this study, the antioxidant activity including $OH^{\bullet}$, ABTS radical scavenging capacity and FRAP of fresh-cut pumpkin during storage was analyzed. As shown in Figure 8a, the $OH^{\bullet}$ radical scavenging capacity increased on day 1 of storage, followed by a gradual decrease, and then increased again. At the end of storage time (6 d), the $OH^{\bullet}$ radical scavenging capacity of slice, strip and piece was 31.65%, 16.19% and 18.15% higher than in whole pumpkin, respectively. Meanwhile, this study found that the cutting operation also promotes the ABTS radical scavenging capacity (Figure 8b) and FRAP (Figure 8c), and wounding intensities have an obvious influence on antioxidant activity in fresh-cut pumpkins. As for ABTS radical scavenging capacity, the levels were ranked as follows: slice > strip > piece. Unlike the ABTS change trend, the order of FRAP levels in fresh-cut pumpkins was slice < strip < piece. In order to explain the mechanism of the enhancement of antioxidant activity in fresh-cut pumpkins induced by wounding stress, the relationship among antioxidant-related indicators was performed.

According to Figure 9, the antioxidant activity (FRAP, $OH^{\bullet}$, ABTS) was significantly positively correlated with antioxidant enzyme (APX, GR) activity, which indicated that the Vc-GSH cycle facilitated the promotion of antioxidant activity in wounded pumpkin. Studies on fresh-cut broccoli [60] and celery [65] also demonstrated this result. Moreover, there was a positive correlation among anthocyanin, GSH, phenol content and antioxidant activity (ABTS, FRAP), whereas there was no correlation among Vc, flavonoid content and ABTS, FRAP. Our findings contradict the previous study on red cabbage [27], in which Vc contributed to improving the antioxidant activity. This was likely due to Vc being decomposed for providing precursors for GSH synthesis in this study. Various studies on fresh-cut carrot [66], celery [67], lettuce [68], broccoli [60], mushroom [69], onion [70], mango [71], pitaya [8], potato [11,72] and red cabbage [28] were consistent with our work which found that the phenol accumulation would further contribute to the increase in antioxidant activity. These results confirmed that the antioxidant compounds play different roles in the process of resisting oxidative damage in fresh-cut products.

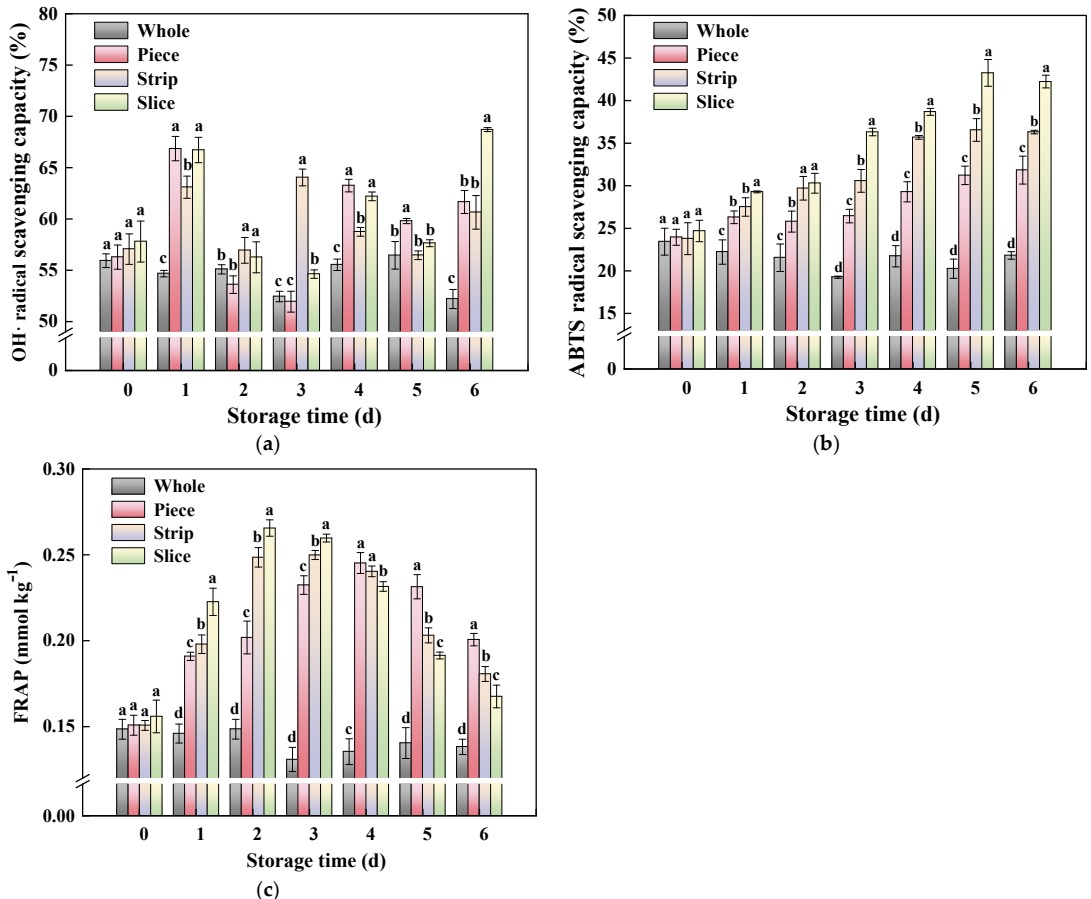

(a)

(b)

(c)

**Figure 8.** Effect of wounding intensities on OH$^\bullet$ (**a**), ABTS (**b**) radical scavenging capacity and FRAP (**c**) of fresh-cut pumpkin during storage at 4 °C for 6 days. Vertical bars represent the standard deviation of the mean (n = 3). Different letters indicate significant differences between the groups (*p* < 0.05).

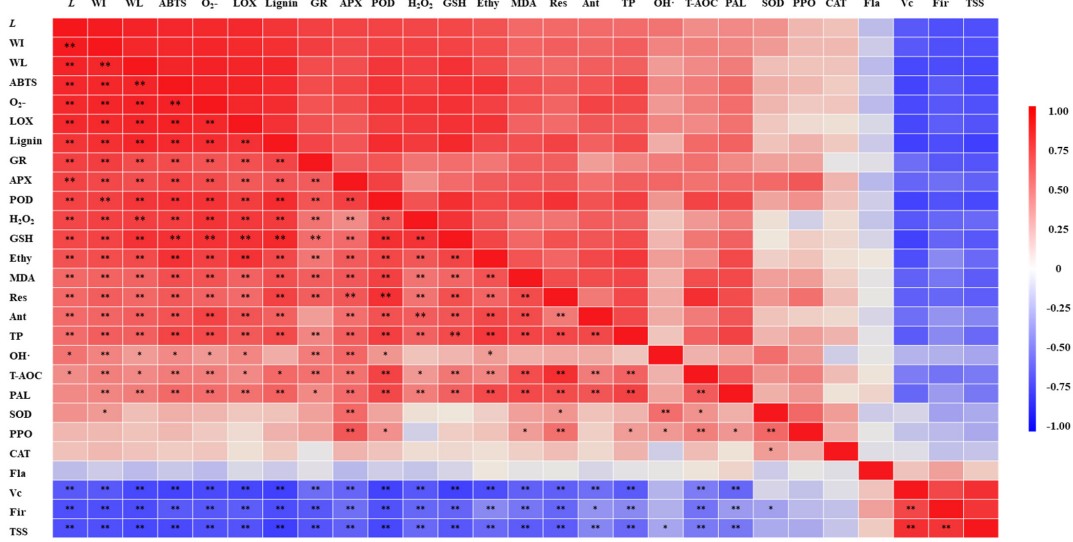

**Figure 9.** Pearson correlation matrix of indexes. * indicates significance at the *p* < 0.05 probability level. ** indicates significance at the *p* < 0.01 probability level. WL, Ethy, Res, Ant, TP, Fla and Fir represent weight loss, ethylene, respiration, anthocyanin, total phenols, flavonoids and firmness, respectively.

## 4. Conclusions

Fresh-cutting operation induced obvious increases in lightness, whiteness index, respiration rate and ethylene content in fresh-cut pumpkin, and there was a significant difference in wounding intensities in these parameters. Meanwhile, wounding stress stimulated the production of ROS, and then improved the process of membrane lipid peroxidation, which resulted in a decrease in appearance, firmness, sensory quality and TSS content. Furthermore, fresh-cutting treatment induced the activity of the critical enzyme involved in phenylpropanoid and ROS metabolism, including PAL, SOD, CAT, APX and GR, resulting in a synthesis of phenols, flavonoids and GSH and further improved the ability to clear ROS, which contributed to the increase in antioxidant activity (OH·, ABTS radical scavenging capacity and FRAP) in wounded pumpkin. These phenomena were more obvious in the cutting method with the enhancement of wounding intensity degree. The collected results revealed that different wounding intensities have an obvious influence on ROS metabolism, Vc-GSH cycle, phenylpropanoid metabolism and membrane lipid peroxidation, which further affect the quality of fresh-cut pumpkin.

**Author Contributions:** Conceptualization, W.H. and Y.G.; Data curation, W.H., Y.G. and Y.W.; Formal analysis, W.H. and Y.G.; Funding acquisition, W.H. and Y.G.; Investigation, W.H. and Y.G.; Methodology, W.H. and Y.G.; Project administration, W.H.; Resources, W.H.; Software, W.H. and Y.G.; Supervision, Y.W. and N.Y.; Visualization, Y.G., Y.W. and N.Y.; Writing—original draft, W.H. and Y.G.; Writing—review and editing, Y.G., Y.W. and N.Y. All authors have read and agreed to the published version of the manuscript.

**Funding:** This research was supported by the "Thirteenth Five-Year Plan" for National Key Research and Development Program (No. 2016YFD0400903), National Natural Science Foundation of China (No. 31471923), National Natural Science Foundation of China (No. 32202120), Zhejiang Agricultural and Forestry University Scientific Research Development Fund Project (Grant No. 2022LFR043).

**Data Availability Statement:** The datasets used and/or analyzed during the current study are available from the corresponding author on reasonable request.

**Conflicts of Interest:** The authors declare no conflict of interest.

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
