# Peer review of "Effect of Wounding Intensity on Edible Quality by Regulating Physiological and ROS Metabolism in Fresh-Cut Pumpkins"

_horticulturae, doi:10.3390/horticulturae9040512_

Round 1

Reviewer 1 Report

Manuscript Number: horticulturae-2322239, titled:

 Effect of Wounding Intensity on Edible Quality by Regulating Physiological and ROS Metabolism in Fresh-cut Pumpkins

Review 1 – 24 March 2023

Dear Editor of Horticulturae

the argument is interesting but it has to be improved. The introduction section has to be better presented and the M&M section has to be completed. The analytical methods have to be described. The bibliography has to be extended, improved and updated. The manuscript is not always arranged as per the instructions for authors of Horticulturae. Inaccuracies in the text.

I suggest a major revision

To the Authors (in detail):

  1. the argument is interesting but it has to be improved. The introduction section has to be better presented and the M&M section has to be completed. The analytical methods have to be described. The bibliography has to be extended, improved and updated. The manuscript is not always arranged as per the instructions for authors of Horticulturae. Inaccuracies in the text.

  1. Introduction section, line 31, please, complete the botanical nomenclature with the abbreviation of the botanist (do not italicize for the abbreviation of the Botanist);
  2. Introduction section, lines 38-40, your have stated that skin residue and other fresh material can be reused but no reference you have included. Please, support your statement with proper references. Find, read and discuss [X1, X2]:

[X1] A Comparative Study on the Nutritional, Antioxidant, Thermal, Morphological and Diffraction Properties of Selected Cucurbit Seeds.

Agronomy 2022, 12 (10) 2242.

https://doi.org/10.3390/agronomy12102242

[X2] Pumpkin Waste as Livestock Feed: Impact on Nutrition and Animal Health and on quality of Meat, Milk, and Egg.

Animals (Basel). 2019 Oct; 9(10): 769.

Published online 2019 Oct 8. doi: 10.3390/ani9100769

  1. sub-section 2.1, no information is given about the origin of pumpkins: geographical area of production; microclimate; greenhouse? fertilizers applied (type, quantity, date);
  2. Sub-section 2.1, no information is given around the degree of ripeness of pumpkins (please do not write: industrial degree, but calculate the days after blossom and the date of picking; weight of pumpkin;
  3. Sub-section-2.1, no information is given around the year of your research;
  4. sub-section 2.1, lines 80, 91 and in the whole manuscript, tables, figures, please, writhe the proper symbol for temperature: °C, please, verify and correct;
  5. Sub-section 3.1, lines 189 and 192, 193 and in the whole manuscript, tables and figures, do not insert a space between L and the asterisk (star): L* and not L*, please, standardize. The same for a* and b*;
  6. Sub-section 3.1 and line 195, do not write in blue color (figure), in addition, please use some recently published paper as a template to verify how figures and tables are reported in the text for Horticulturae;
  7. Sub-section 3.1, line 186 (colourimetric), line 210 (color), please, decide for the whole manuscript if to use the American or the British spelling. Standardize;
  8. Sub-section 3.2 (p<0.05); sub-section 2.10 (p ≤ 0.05): decide if to include spacing between letter, symbol and numeric value, please be consistent, standardize in the whole manuscript, captions of tables and figures. In addition, verify if <0.05 or ≤0.05;
  9. sub sub-section 3.3.4, before to discuss your data, explain that ABTS and FRAP are very important assays and that they are applied to many vegetables matrices and support this statement with proper references [X3, X4, X5]:

[X3] Nutritional quality, mineral and antioxidant content in lettuce afected by interaction of light intensity and nutrient solution concentration.

Scientific Reports | (2020) 10:2796 | https://doi.org/10.1038/s41598-020-59574-3

[X4] Genetic analysis and interaction among CUPRAC, FRAP, phytochemical and phenotypic traits in cauliflower (Brassica oleracea var. botrytis L.).

International Journal of Chemical Studies 2019; 7(1): 1484-1494

[X5] The Impact of the Method Extraction and Different Carrot Variety on the Carotenoid Profile, Total Phenolic Content and Antioxidant Properties of Juices.

Plants 20209(12), 1759; https://doi.org/10.3390/plants9121759

  1. Results and discussion section, before to discuss about Flavanoid content, explain that flavonoids are studied in many matrices and support this statement with proper references [X6, X7]:

 [X6] Physico-chemical Stability of Blood Orange Juice during Frozen Storage.

International Journal of Food Properties 20:sup2, 1930-1943 (2017). https://doi.org/10.1080/10942912.2017.1359184

[X7] Bioactive compounds and biological activity of ginger.

J. Multidiscip. Sci. 2019, 1(1), 1-7.

  1. In the caption of figure 9, when you have indicated the significance, you have not italicised p, whereas in many cases you have italicised. Please standardize in the whole manuscript;

  1. References section, please, be consistent with the instructions for authors of Horticulturae: the publication year has to be written in bold and the volume has to be italicised;

  1. Please, write in blue color or evidence differently the corrections you will do.

I suggest a major revision

Regards.

Author Response

Please see the attachment, thank you very much!

Reviewer 2 Report

In the presented manuscript, the authors examined the impact of injury intensity on the functional and physiological parameters of fresh pumpkin. I was very interested in the article. The introduction was written correctly and clearly. It is completed with a goal. The methodology is very accurate, a big plus for the fact that you can certainly repeat the experiments. The presentation of the results is very good and legible. The discussion refers to the presented results. The selection of literature is correct, although many of the literature items are old (in the future, please search for as many items as possible from the last 5 years).

Here are some minor comments for improvement.

Line 88. Please explain why cucumber was chosen as the control sample? Even though all of these species are gourds, wouldn't zucchini be better?

Line 92. Please write more information about color measurements: device, illuminant, observation angle, number of reading per sample, aperture size. What standard was used to calibrate?

Line 104. What is the conversion factor 1.96? Please explain.

Line 120, 124, 130, 141. What is this plant kit? Please provide more information.

Line 148. Please use "×" to record the spin.

Line 155. What commercially available kit? Details please.

Table 1. Values would be easier to read if statistical significances were marked in superscript, for example 16.00±0.54 a

What are the differences in the content of flavonoids and anthocyanins over time? Sometimes they decrease, sometimes they increase?

Figure 4 is incorrectly signed.

In figure 6a the values are not combined. Is that how it's supposed to be?

Author Response

(The authors gave the same response as above.)

Round 2

Reviewer 1 Report

Manuscript Number: horticulturae-2322239, titled:

Effect of Wounding Intensity on Edible Quality by Regulating Physiological and ROS Metabolism in Fresh-cut Pumpkins

Review 2 – 11 April 2023

Dear Editor of Horticulturae

the Authors have included the large part of my comments, anyway some improvement more is necessary. The scientific names have to be written in light of the International binomial nomenclature: Genus in capital letter, Genus and species have to be italicised.

I suggest a minor revision

To Authors (in detail):

  1. Sub-section 2.5, lines 141-145 and 149-150, do not write here the whole HyperText Transfer Protocol (HTTP) address but insert one number between brackets and include it in the references section;
  2. Sub-section 2.6, lines 156-156, do not write here the whole HyperText Transfer Protocol (HTTP) address but insert one number between brackets and include it in the references section;
  3. Sub-section 2.7, lines 170-174, do not write here the whole HyperText Transfer Protocol (HTTP) address but insert one number between brackets and include it in the references section;
  4. Sub-section 2.8, lines 189-195, do not write here the whole HyperText Transfer Protocol (HTTP) addresses but insert one number between brackets and include them in the references section, including the day of the access;
  5. Sub sub-section 3.2.2., line 323, please, include between the same two brackets the references number: [33-34] and not [33] [34];
  6. References section, refs 33-34,54-55-56 and in the whole section, the journal name has to be italicised;
  7. Ref 34, the name of the author is: Mele, M.A.;
  8. Ref 45, the scientific name has to be italicised: Brassica rapa;
  9. Ref 46, the abbreviated title is: Physiol. Plant.;
  10. Ref 50 and in the whole section and manuscript: the scientific name has to be italicised;
  11. Ref.53, Botrytis cinerea (italicised and the Genus in capital letter);
  12. Ref 55, the scientific name has to be italicised;
  13. Ref 57: Martinez-Hernandez, B.G.;
  14. Ref.62: the correct scientific name is Agaricus bisporus: the Genus in capital letter and genus and species have to be italicised;
  15. Ref 64: Robles-Sanchez, M.R.;
  16. Please, write in blue color or evidence differently the corrections you will do.

I suggest a minor revision

Regards.

Author Response

(The authors gave the same response as above.)
